

# Planetary boundary layer evolution over the Amazon rain forest in episodes of deep moist convection at ATTO

Maurício I. Oliveira[1,a], Otávio C. Acevedo[1], Matthias Sörgel[2], Ernani L. Nascimento[1], Antonio O. Manzi[3], Pablo E. S. Oliveira[1], Daiane V. Brondani[1], Anywhere Tsokankunku[2], and Meinrat O. Andreae[2,4]

[1]Universidade Federal Santa Maria, Departamento de Física, Santa Maria, RS, Brazil
[a]now at: Center for Analysis and Prediction of Storms, and School of Meteorology, University of Oklahoma, Norman, Oklahoma.
[2]Max Planck Institute for Chemistry, P.O. Box 3060, 55020, Mainz, Germany
[3]Instituto Nacional de Pesquisas da Amazônia (INPA), Clima e Ambiente (CLIAMB), Av. André Araújo 2936, Manaus-AM, CEP 69083-000, Brazil
[4]Scripps Institution of Oceanography, University of California San Diego, La Jolla, USA

*Correspondence to:* Maurício I. Oliveira (mauricio.meteorologia@gmail.com)

**Abstract.** In this study, high-frequency, multi-level measurements performed from late October to mid-November of 2015 at a 80-m tall tower of the Amazon Tall Tower Observatory (ATTO) project in central Amazonas State, Brazil, were used to diagnose the evolution of thermodynamic and kinematic variables as well as scalar fluxes during the passage of outflows generated by deep moist convection (DMC). Outflow associated with DMC activity over or near the tall tower was identified

5  through the analysis of storm echoes in base reflectivity data from S-band weather radar at Manaus, combined with the detection of gust fronts and cold pools utilizing tower data. Four outflow events were selected, three of which took place during the early evening transition or nighttime hours and one during the early afternoon. Results show that the magnitude of the drop in virtual potential temperature and changes in wind velocity during outflow passages vary according to the type, organization, and life cycle of the convective storm. Overall, the nocturnal events highlighted the passage of well-defined gust fronts with moderate

10  decrease in virtual potential temperature and increase in wind speed. The early afternoon event lacked a sharp gust front and only a gradual drop in virtual potential temperature was observed, probably because of weak or undeveloped outflow. Sensible heat flux ($H$) experienced an increase at the time of gust front arrival, which was possibly due to sinking of colder air. This was followed by a prolonged period of negative $H$, associated with enhanced nocturnal negative $H$ in the storms' wake. In turn, increased latent heat flux ($LE$) was observed following the gust front, owing to drier air coming from the outflow; however,

15  malfunctioning of the moisture sensors during rain precluded a better assessment of this variable. Substantial enhancements of Turbulent Kinetic Energy (TKE) were observed during and after gust front passage, with values comparable to those measured in grass fire experiments, evidencing the highly turbulent character of convective outflows. The early afternoon event displayed slight decreases in the aforementioned quantities in the passage of the outflow. Finally, a conceptual model of the time evolution of $H$ in nocturnal convective outflows observed at the tower site is presented.



## 1  Introduction

Deep moist convection (DMC) is a ubiquitous feature of the atmospheric environment of the Amazon rain forest. Because of intense diurnal solar heating in the moist planetary boundary layer (PBL), conditional instability builds up and convective storms form regularly in order to redistribute energy in the atmospheric column (Johnson and Mapes, 2001). The barotropic
regime of the Amazonian atmosphere, devoid of strong vertical wind shear, most often gives rise to convective storms that display a life cycle typical of single cells (or "pulse-type storms") described in Byers and Braham (1949). During the stage at which convective storms produce precipitation, latent cooling from the evaporation of rain (or melting of ice species below the $0°C$ isotherm) cools air parcels that eventually become negatively buoyant. The acquired downward acceleration is reinforced by the drag caused by hydrometeor loading, and a downdraft is initiated (Wakimoto, 2001). Downdrafts, in turn, introduce
cooler and drier air from above the PBL (and cloud base) into the surface. Since this airflow has different thermodynamic and kinematic properties than the near-surface air mass, it disturbs the mean evolution of the PBL.

Much of the knowledge on the effects of DMC on PBL evolution has been gained from research based on the GARP Atlantic Tropical Experiment (GATE) in 1974 (Kuettner and Parker, 1976). Fitzjarrald and Garstang (1981, hereafter referred to as FG81), using Boundary Layer Instrument System (BLIS) profiles collected on three ships, showed that DMC can affect
drastically the thermodynamic evolution of the oceanic mixed layer (ML) near the Intertropical Convergence Zone (ITCZ). They showed for a squall line event that convective downdrafts modify the evolution of the ML primarily by inducing an abrupt drop in temperature, usually accompanied by a drop in moisture, resulting in shallower MLs. After this stage, the cooler and drier ML is maintained by the continued influence of downdrafts. Finally, they identified a recovery phase in which the ML returns to its undisturbed state. This stage initiates in the wake of the storms and may take 7-10 h for the ML to re-establish
undisturbed conditions. Studying the same squall line system analyzed by FG81, Johnson and Nicholls (1983) provided a composite analysis of all marine rawinsonde observations that were collected during the event. They found similar reduction in the mixing layer height following the passage of the squall line, with associated temperature and moisture drops of $4°C$ and 3-4 g kg$^{-1}$, respectively.

The occurrence of DMC has significant impacts upon the evolution of surface scalar fluxes, since convective outflows are
responsible for cooling and drying the PBL (Fitzjarrald and Garstang, 1981; Johnson and Nicholls, 1983; Saxen and Rutledge, 1998). Johnson and Nicholls (1983) computed sensible and latent heat fluxes over an area encompassing the entire convective system and surrounding areas. The authors found large sensible and latent heat flux enhancements in the wake of the squall line, which increased, respectively, by factors of 5 and 2 over their undisturbed values of 10 W m$^{-2}$ and 90 W m$^{-2}$. Scalar flux enhancements in DMC situations was further investigated by Saxen and Rutledge (1998, henceforth, SR98), who computed
surface fluxes from meteorological data measured by an instrumented buoy as part of the Coupled Ocean-Atmosphere Response Experiment (COARE) of the Tropical Ocean Global Atmosphere (TOGA) project that took place in the western Pacific warm pool from November 1992 through March 1993. They found latent and sensible heat flux enhancements reaching peak values of 60 W m$^{-2}$ and 250 W m$^{-2}$ for large, organized Mesoscale Convective Systems (MCSs). For less organized storm types, such as maturing linear MCSs and scattered storms, weaker heat fluxes were reported.



While most studies have focused on the DMC-PBL interaction over the tropical oceans, the evolution of turbulent fluxes in DMC situations in forest environments has also been addressed, either observationally (Fitzjarrald et al., 1990; Betts et al., 2002; Gerken et al., 2016) or numerically (Garrett, 1982). Specifically for the Amazon rain forest, Fitzjarrald et al. (1990) showed that, in daytime conditions, outflow air penetrates the weakly-stratified layer above canopy level and, depending on the
strength of downdrafts, can occasionally penetrate the semi-permanent stable layer within the canopy, leading to deep mixing throughout the inside and above the forest. To further justify the relevance of better documenting the DMC effects on the Amazonian PBL it is important to recognize that it can also affect forest-atmosphere exchanges of chemical species. Even though turbulence is reduced below canopy, the perturbation induced by outflows may engender the venting of hydrocarbons and trace gases out of the canopy (Fitzjarrald et al., 1990; Fuentes et al., 2016). In addition, outflows can promote sudden increases of
ozone concentration in the PBL through downward transport of mid-tropospheric ozone-rich air by storm downdrafts (Betts et al., 2002; Gerken et al., 2016).

In this context, high-frequency tower measurements performed at the Amazon Tall Tower Observatory (ATTO) (Andreae et al., 2015) experiment site provide an excellent way to assess the impacts of tropical DMC on the mean evolution of the PBL in the Amazon rain forest. In this study, we employ multiple-level high-frequency measurements performed at one of the
ATTO towers situated in central Amazonas State, Brazil, in tandem with radiosonde and Doppler radar data, to carry out a multiplatform analysis of the effects of DMC on the evolution of turbulent quantities in the lower Amazon PBL. Differently from previous studies, which focused mainly on daytime changes in the ML caused by storms over tropical oceans, most of our results are from storm events that occurred during nighttime hours when the establishment of stable boundary layers was either underway or already present. Furthermore, previous studies of DMC-PBL have devoted little attention to the evolution
of turbulence intensity in observed tropical DMC events, partially because of a lack of high-frequency micrometeorological measurements in storm situations. In light of this need, we take advantage of the high-frequency tower observations from ATTO to present the evolution of turbulent kinetic energy (TKE) in storm outflows.

This paper is organized as follows: Section 2 provides information about the datasets employed in this study along with the methods utilized for identifying storm events and computing turbulent quantities at the tower. Section 3 presents an overview
of the main aspects of the convective storms that were analyzed, in terms of radar features and meteorological changes detected at the tower. Section 4 is aimed at investigating the mechanisms by which the fluxes of sensible and latent heat are enhanced throughout the instrumented tower depth and how they relate to PBL evolution in the wake of storms. In Section 5 we investigate the TKE evolution during storm outflows using high-frequency tower observations from ATTO. Finally, the conclusions are presented in Section 6.

## 2 Data and Methods

### 2.1 ATTO data and instrumentation site

The primary data source employed in this investigation consists of high-frequency (10 Hz) micrometeorological measurements performed at the 80 m tall walk-up tower, located 150 km northeast of Manaus, in the Uatumã Sustainable Development





Reserve. The tower is situated at a base elevation of 130 m above sea level (a.s.l.). A detailed description of the site, instrumentation capabilities, underlying vegetation and nearby topography, as well as other relevant features, can be found in Andreae et al. (2015). The study period extended from 29 October 2015 to 20 November 2015. Data used here were recorded at five distinct height levels, namely: 14, 22, 41, 55 and 81 m (above ground level; a.g.l.). The average height of trees in this portion

of the Amazon rain forest is approximately 37 m (Andreae et al., 2015). Therefore, the first two measurement levels reside within the forest canopy, while the three uppermost levels are situated above it. At 14, 41 and 55 m, sonic anemometers (model CSAT3, Campbell Scientific, Inc.) performed fast response wind ($u$, $v$, and $w$) measurements in addition to sonic virtual temperature ($T_v$). Using different instrumentation, similar temperature and wind measurements were obtained at 22 m (Irgason, Campbell Scientific Inc.) and at 80 m (Windmaster, Gill Instruments Limited) anemometers.

Computation of turbulent quantities from tower data such as mean flow, heat fluxes and turbulent kinetic energy were accomplished by employing Reynolds averaging at 1-min time intervals. We have chosen such a short time window primarily because of the nonstationary nature of the events under study, but also to avoid contamination from low-frequency, non-turbulent processes, and, therefore, guarantee that the discussion refers to turbulent quantities alone. Most of the cases analyzed occurred when stable stratification was present at the site. This choice was based on the results of Campos et al. (2009), who

found that the time scale for turbulent fluxes at nighttime was consistently smaller than 200 s above a similar Amazonian canopy.

## 2.2 Doppler radar data

Radar data used in this study came from the operational S-Band Doppler radar located in Manaus (3° 09′ S; 59° 59′ W), operated by the Department of Airspace Control (SIPAM/DECEA; acronym in Portuguese) of the Brazilian Air Force. The

Manaus radar is a single polarization system with a relatively broad beamwidth of approximately 1.8°. In short pulse mode, the radar operates with a range and pulse repetition frequency of 250 km and 600 Hz, respectively. In volume scan mode, the radar performs a full set of plan position indicators (PPIs) at 15 elevations at 10 min intervals. Radar data were plotted using the Python ARM Radar Toolkit (Py-ART) software (Helmus and Collis, 2016).

## 2.3 Selection of DMC events

In this study, the selection of DMC events was accomplished by following a two-step procedure relying on Doppler radar imagery and thermodynamic and kinematic changes associated with the storms as detected at the ATTO walk-up tower. The first step consisted of subjectively inspecting radar reflectivity fields using low-elevation Plan Position Indicator (PPI) to identify the passage of convective storms over or near the instrumentation site. Only storms that produced detectable impacts on the evolution of meteorological variables at the tower site were selected. To that end, time series of virtual potential temperature

($\theta_v$) and horizontal wind speed ($V_h$) measured at levels above the forest canopy were analyzed to identify thermodynamic and kinematic changes caused by gusts fronts (i.e., low-level outflow) from the convective storms.

In Addis et al. (1984), gust fronts were detected by imposing a minimum virtual temperature decrease of 0.5°C on 3-min-averaged data from BLIS data. However, in this study we have chosen not to apply any threshold to $\theta_v$ or $V_h$ variations to





detect a storm outflow, but simply to subjectively select those events that displayed noticeable perturbations in the temperature and wind fields at the time of storm occurrence. This choice was motivated by: (a) our interest in evaluating DMC events that influenced the evolution of PBL properties through their outflows, regardless of the magnitude of the temperature and wind variations produced by them; (b) recognizing that perturbations associated with convective storms were easy to identify as they represented drastic interference in the mean evolution of PBL quantities; (c) the short period of study, which did not demand defining a set of objective criteria as would be the case for large datasets as in Addis et al. (1984). We should add that when a convective event consisted of more than one cell affecting the tower, the entire period of DMC activity was investigated in order to obtain the most complete description of PBL evolution during the full life cycle of the storm system.

Following the aforementioned procedure, four DMC events were selected for investigation. Dates, duration, radar characteristics, and other relevant features of the storm events are presented in Table 1. The wind speed increase in the outflow (i.e., the gust front intensity) was measured as the bulk difference between the 1-min mean wind in the pre-storm environment and the maximum wind after storm arrival. Similarly, the maximum temperature drop (i.e., the cold pool intensity) was measured as the bulk difference between the 1-min mean $\theta_v$ in the pre-storm environment and the minimum $\theta_v$ obtained after outflow establishment at the tower region. Note that the times of maximum wind increases and temperature drops may not coincide as the largest wind increases usually occur just as the leading edge (i.e., the gust front) impacts the tower and the largest temperature deficits often occur after the storm's cold pool is well established.

## 3 Overview of the DMC events

Previous studies of DMC-PBL interaction have demonstrated the importance of characterizing morphological aspects of the convective activity that disturbs the PBL with the aid of radar imagery. For example, SR98 classified storm organization according to the horizontal extent of precipitation echoes in the reflectivity field, the presence (or absence) of stratiform precipitation, and whether convection was linearly organized or not. In this study, all of the four storm events investigated consisted of either a single cell or a small cluster of multicell storms (Moller et al., 1994). These storms never developed upscale to reach the minimum horizontal extent of 100 km necessary to fit the classification of a Mesoscale Convective System (MCS) (Houze, 2014). In comparison to SR98, the storms on 31 October (event 1), 2 November (event 2), and 4 November (event 3) mostly resembled the unorganized arrangement that they referred to as sub-MCS-scale nonlinear systems. The exception was event 4, on 9 November, which displayed a more organized structure, but remained slightly below the minimum spatial threshold for MCS classification.

Storms struck the walk-up tower site at different times of the day. Two events (1 and 2) took place during the late afternoon or early evening transition (EET) while event 3 occurred during late morning hours. Event 4 occurred at dawn and was the longest-lived event. These differences in the time of storm occurrence are relevant as the convective outflows interact with the PBL during distinct stages of its evolution. In the following subsections, a description of each event is presented, focusing on their radar characteristics and the intensity of the thermodynamic and kinematic effects detected at the walk-up tower site.



### 3.1   31 October 2015 - Line of multicells (Event 1)

At approximately 17:15 LST on 31 October 2015, the Manaus Doppler radar indicated a northeast-southwest-oriented band of convective cells advancing over the eastern-northeastern Amazon as part of a larger area of intense but disorganized convective activity (Fig. 1a). At the southern tip of the convective band, westward-moving decaying cells merged with semi-stationary cells to the south of the ATTO site and started moving northwestward. As the cells passed directly above the site, they intensified as noted by a rapid increase in reflectivity.

Pre-storm measurements of winds and virtual potential temperature at the walk-up tower revealed a slight tendency of decreasing turbulence and temperature typical of pre-sunset conditions. However, as the outflow from the storm cluster arrived at ATTO, a sudden drop in $\theta_v$ was observed in tandem with an increase in wind speed at heights above the canopy (Fig. 2a and Fig. 3a). The temperature and wind disturbances were significantly damped inside the canopy, at 14 m and 22 m heights. This is expected since the dense rain forest and its interior stable layer act to inhibit strong air flow (Fitzjarrald et al., 1990). The flow remained very turbulent and $\theta_v$ continued to decrease, though at a slower rate, until 17:35 LST when a new 4 K $\theta_v$ drop was observed at the same time a $V_h$ increase was observed. This cold-air reinforcement was probably caused by a secondary outflow surge trailing the leading gust front; in fact, the 17:33 LST reflectivity image (not shown) displayed a brief period of convective re-intensification, preceding the decay of the system and onset of stratiform precipitation. The PBL cooling and consequent stabilization induced by this storm system during the early evening period caused an effective "early nightfall" as described by Fitzjarrald et al. (1990). In this situation, the establishment of a shallow, cool near-surface stable layer occurs earlier than it would be the case for a typical undisturbed diurnal cycle.

After the strongest cooling associated with the convective active stage of the system, minimum $\theta_v$ values (299-300 K) were attained by 18:20 LST and low-amplitude $\theta_v$ perturbations persisted in the wake of the storm as result of weaker downdrafts. Once the perturbation caused by the storms decayed, the $\theta_v$ time series showed that a steady state was attained, though it took place at much lower temperatures than in the undisturbed pre-storm environment. Full PBL recovery did not occur for this EET event, since solar heating had long ceased and surface stabilization (and thus, demise of the ML) was underway when storms impacted the tower site.

### 3.2   2 November 2015 - Isolated cell (Event 2)

Small clusters of short-lived thunderstorms were observed by the Manaus radar near the ATTO location during the late afternoon and early evening period on 2 November 2015 (Fig. 1b). Around 18:00 LST, the gust front from an isolated short-lived cell reached the tower resulting in maximum 6-8 K $\theta_v$ drops and an attendant increase in $V_h$ as high as 6 m s$^{-1}$ for all above-canopy levels (Fig. 2b and Fig. 3b). Interestingly, a short increase in $\theta_e$ occurred associated with a 1 g kg$^{-1}$ increase in water vapor mixing ratio ($r_v$) briefly after the gust front arrival (not shown). As the gust front impacted the tower after sunset, an early nightfall effect was also observed, similar to event 1.

The convectively active stage of the storm over the tower lasted approximately 20 min, being considerably less than what was observed with event 1, which was associated with a much larger storm and more easily detected in the $\theta_v$ times series.



During the disturbed period, a wave-like behavior could be seen in both temperature and wind time series throughout the whole profile suggesting the presence of large eddies capable of penetrating deep inside the forest. Different from event 1, a recovery phase did exist for this event, despite solar heating having already ceased.

There are some clear differences between the recovery phase inside and above the canopy, as indicated by in-canopy mea-
surements. The levels above the canopy show full recovery after 50 min, as stated above, while below the canopy, cooler temperatures are maintained long after the above-forest air mass had attained a new steady state. In this scenario, it seems that the forest slowed down the recovery in its interior, thus fostering the establishment of a very stable stratification next to the ground. Hence, some process(es) related to upward fluxes of heat and moisture must have occurred in order to warm and moisten the layers near the top and above the canopy. The mechanisms responsible for these processes will be described in
detail in Section 3.

### 3.3  4 November 2015 - Scattered cells (Event 3)

Around 10:20 LST, an unorganized cluster of convective cells rapidly formed around the ATTO site at the back side of a westward-moving MCS (Fig. 1c). $\theta_v$ gradually began to decrease from 305.5 K at 10:48 LST; surprisingly, this drop in $\theta_v$ was followed by only a modest increase in $V_h$. Wind speed slightly increased from 6 m s$^{-1}$ up to 8 m s$^{-1}$ and then weakened when
a minimum $\theta_v$ of 302 K was reached at 11:27 LST, amounting to a total 3.5 K decrease (Fig. 2c and Fig. 3c).

This event clearly displayed a behavior that was quite different from the other cases studied, especially in light of the slow nature of the potential temperature (wind speed) decreases (increases). The most probable explanation for this anomalous behavior is that the arrival of the outflow from the scattered storms at the ATTO site was not preceded by a sharp gust front, as was the case for the other events. Rather, it is plausible that merging of weak outflows from the incipient or decaying storm
cells generated a slow-moving cold pool that gently spread over the site. If this was the case, it is safe to state that, although the PBL was disturbed by the DMC outflow, the downdraft cores of the parent cells or the strongest portion of their gust fronts did not pass directly over the instruments.

### 3.4  9 November 2015 - Strong cluster with trailing stratiform precipitation (Event 4)

The most organized system investigated was associated with a large southwestward moving cluster of strong storms with a
trailing stratiform precipitation region (Fig. 1d). Before the arrival of the storm system, a sequence of smaller cells advanced over the tower site, producing a weak wave-like perturbation in both $\theta_v$ and $V_h$ (Fig. 2d and Fig. 3d). The weakness of these disturbances is probably associated with the existence of a well-established nocturnal stable boundary layer (SBL). It is well known that SBLs tend to damp convective downdrafts (Market et al., 2017) and therefore, the weak downdrafts from the small cells were unable to drastically disturb the SBL. At 04:00 LST, however, the large cluster of cells passed by the tower causing
a $\theta_v$ 3 K drop and winds increasing from 1 m s$^{-1}$ to 10 m s$^{-1}$, making this event the strongest one in terms of gust front strength. The outflow from the system was strong enough to penetrate the in-canopy stable layer, even in the presence of the aforementioned deeper SBL. It is suggested that this event contained the most intense downdrafts among the four cases.





Considering only the main convective system, the convectively active period in this episode was also longer compared to the other events. An "attempt" of a recovery phase was observed as a slight increase in $\theta_v$ around 04:00; nonetheless, it was short-lived (lasting approximately 15 min) owing to the existence of trailing stratiform precipitation (with embedded weaker echoes) following the storm system. The persistence of the DMC over the ATTO site reduced the early morning incidence of
solar radiation and slowed down the subsequent development of a ML.

## 4  Evolution of heat fluxes and intensity of outflow turbulence

### 4.1  Sensible heat flux ($H$)

Surface heat fluxes play a major role in the initiation process of convective storms in tropical regions, as intense diurnal heating drives thermals or plumes that grow upscale into large cumulonimbus clouds. On the other hand, when convective downdrafts
introduce cool air from aloft into the PBL, the evolution of surface heat fluxes may also be affected significantly. Figure 4 displays the evolution of the sensible heat flux ($H$) measured at the tower for the four DMC events. For the sake of consistency, we shall first discuss overall similarities in the behavior of $H$ for events 1, 2, and 4, separately from event 3, before scrutinizing the particular characteristics of each event. We will address event 3 separately, because, as discussed in Section 3, it displayed a quite distinct behavior from the other three events.

Prior to the occurrences of events 1, 2, and 4 (Fig. 4a, b, and d), $H$ was downward (negative), as the boundary layer at that time had already experienced the evening transition. A common feature in these three events was an abrupt switch to upward $H$ as soon as the gust front arrived at the tower, especially for levels above the canopy. Inside the canopy, positive $H$ occurred, but it was weaker than above the canopy. Peak $H$ values were most pronounced during the gust front phase and arrival of the storm, with 1-min mean values exceeding 300 W m$^{-2}$, 175 W m$^{-2}$, and 150 W m$^{-2}$ for events 1, 2, and 4, respectively. These $H$
enhancements agree with findings from previous studies that showed an increase in $H$ following gust front passages for marine DMC (Johnson and Nicholls, 1983; Fitzjarrald et al., 1990; Saxen and Rutledge, 1998); however, as discussed below, it seems that the mechanisms responsible for the upward H found here differ from those governing daytime DMC-ocean interactions.

During this early stage of the DMC activity over the tower, the $H$ time series is well correlated with V$_h$ (as well as with w; not shown), with peaks in wind speed matching peaks in $H$. This is particularly evident in the double-peak structures of both
V$_h$ and positive $H$ for event 2. Combined with the fact that the Manaus radar showed the strongest echo situated over the tower site at this time (not shown), this behavior indicates that $H$ enhancements are related to processes associated with the arrival of the gust front at the tower location. In fact, an analysis of the time series of w$'$ and $\theta_v$ shows that both variables are mostly negatively skewed at the gust front arrival and that this period is marked by a strong correlation between these two quantities. Thus, it seems that the arrival of the gust front and its associated convective downdraft during nighttime conditions resulted
in enhanced $H$ through intense turbulent mixing of cool air from above the PBL. The analysis of the temperature time series along the events (Fig. 5a, b, and d) shows that the short period with upward $H$ coincides with a brief inversion of the vertical temperature gradient, an interval when the temperature within the canopy (26 m) is larger than that above it, characterizing an unstable layer at just above the surface. In fact, a stable layer was already established when the event took place but, as the cold





air moved down, there was a brief period when the thermal gradient switched sign, characterizing an unstable layer, at exactly the period when the upward $H$ occurred.

Soon after the most intense DMC perturbations stage ends, a sudden transition to a prolonged period of negative $H$ begins. This period is associated with the establishment of the trailing precipitation zone and windy wake of the convective system,
which usually persist for tens of minutes to hours after the core of the storm has passed. Within the period of negative $H$, minimum 1-min mean values up to -350 W m$^{-2}$, -800 W m$^{-2}$, and -200 W m$^{-2}$ were observed in events 1, 2, and 4, respectively. In event 4, in particular, short periods of weak negative $H$ occurred in association with small, short-lived cells (Section 3) within the larger storm system. Such a period of markedly negative $H$ has not been addressed in prior studies owing to the fact that they were mainly interested in daytime convective storms interacting with well-established MLs. Fitzjarrald et al. (1990)
mentioned the existence of negative kinematic sensible heat flux values during their daytime DMC cases in the Amazon but did not provide an in-depth discussion of the reasons for these negative fluxes.

The strong persistent negative $H$ period coincides with the continued DMC perturbation as evidenced by lowered temperature, strong winds and turbulence, similarly to the period of positive $H$. During EET (events 1 and 2) or nighttime conditions (event 4), where a SBL is forming or is already established, the effect of convective outflows seems to enhance pre-existing
negative $H$ through cooling and increased turbulence by strong winds, with the wind component of the perturbations being the main modulator of the duration of negative $H$ enhancements.

In summary, positive, intense $H$ enhancements are primarily a feature of the convectively active phase of the storm system, i. e., the arrival of the gust front, intense downdrafts and the brief formation of an unstable layer as the air travels downward. In turn, after the strongest part of the convective system moved away from the tower and weaker downdrafts and windy surface
conditions remain, $H$ rapidly becomes negative again (as it was in the undisturbed conditions), but displays higher values for several minutes owing to continued higher surface wind and lower temperature.

We now turn our attention to event 3 (Fig. 4c). As shown in Section 3.3, event 3 was the only one to occur during daytime hours, under strong insolation and mixing. Under these conditions, increasing $H$ values were in place by mid-morning, when $\theta_v$ started to decrease at 10:20 LST. As the temperature gradually dropped, a negative tendency in $H$ was observed, especially
above the canopy. At the tower levels above the canopy, even negative values (approximately -50 W m$^{-2}$) were observed, while within the canopy $H$ did not change considerably (although small perturbations were noticed). After 13:00 LST, there was a tendency of slow warming and weakening winds.

## 4.2 Latent heat flux ($LE$)

Latent heat fluxes are mainly controlled by the wind magnitude and vertical humidity gradients. SR98 showed that large
enhancements of $LE$ occur in the wake of oceanic gust fronts, which can sometimes be over 300% stronger than in pre-storm conditions. Time series of $LE$ for our case studies are shown in Fig. 6; because no water vapor measurements were available at 80 m for event 4, the only full time series depicted for this event is for the 22 m height. Time series of $LE$ exhibited an appreciably noisier behavior than their $H$ counterparts. For this reason, the analysis of $LE$ will be more qualitative, with less focus on fine details in the magnitude of the fluxes. Figure 6a displays abrupt $LE$ enhancements taking place as the gust fronts





arrived at the tower. Most of the DMC-disturbed period was marked by positive *LE* enhancements, contributing to a net positive *LE* at 80 m.

The occurrence of such enhancements of *LE* as a response to convective storm downdrafts has been demonstrated in previous studies (Johnson and Nicholls, 1983). Mixing ratio deficits ranging from 4-6 g kg$^{-1}$ were observed at 80 m, in line with results

from other studies. It is worth noting that significant surface drying was observed in our events, regardless of the size of the storm. As an example, consider storm event 2 (small single cell storm); even though the horizontal dimension of this storm was small compared to the other cases (especially events 1 and 4), its downdrafts were able to bring down air from sufficiently high altitudes to produce significant surface drying. Thus, contrary to SR98, who showed that large storm systems (MCSs) are more prolific in drying the PBL through mesoscale downdrafts, we show that tropical isolated convection can also be able to

produce intense PBL drying, as long as it can develop deep, virtually undiluted downdrafts.

## 5 Turbulence intensity of convective outflow

Cool outflows from convective storms tend to be very turbulent in nature. As discussed throughout the paper, many studies have shown significant enhancements in turbulent quantities, such as heat, moisture and momentum fluxes, during and after the occurrence of convective outflows (Johnson and Nicholls, 1983; Fitzjarrald et al., 1990; Saxen and Rutledge, 1998). For

example, in a high-resolution numerical study addressing some of the shortcomings of utilizing Large Eddy Simulations for severe storms research, Markowski and Bryan (2016) provided evidence of the abundance of storm-generated turbulent eddies within the outflow (see their Fig. 1).

Probably, the simplest way to analyze the intensity of turbulence in a given flow is to compute the Turbulent Kinetic Energy (TKE) associated with it (Stull, 1988). However, previous studies investigating PBL processes have typically employed dif-

ferent quantities to assess turbulence intensity (e.g., the standard deviation of vertical velocity, Acevedo et al. (2009); Thomas et al. (2013)). In this study, we opted to compute TKE over other quantities because of the simplicity in directly interpreting the underlying physics of energy changes associated with momentum transfers in convective outflows.

Figure 7 shows time series of TKE for the four storm events investigated here. As also found with heat fluxes, events 1, 2, and 4 displayed sudden increases in TKE as soon as the gust fronts arrived at the tower. TKE rises to very high values,

exceeding 8 m$^2$ s$^{-2}$ at the time of the most intense downdraft in event 4, for example. These values are much larger than those observed in typical undisturbed PBL situations, being comparable in magnitude to TKE reported during grassfires (Clements et al., 2008).

TKE peaks follow closely those seen in the V$_h$ time series, as is expected since stronger winds imply augmented mechanical (i.e., shear) production of turbulence. Although the computation of the forcing terms in the prognostic TKE equation is outside

the scope of this paper, it is reasonable to infer that one very important forcing mechanism of turbulence within the storm outflow is the mechanical production. Turbulence production by buoyancy, in contrast, is an unlikely mechanism here since storm-induced temperature drops and the nighttime character of the events would point to buoyancy sinks and turbulence destruction. However, we cannot dismiss the role played by turbulence transport and pressure correlation terms to TKE evolution





in these outflows. To assess these processes, it would be necessary to conduct an in-depth qualitative analysis of each term in the prognostic TKE equation, a topic that will be addressed in a future study.

Considering event 3 (Fig. 7c), the evolution of TKE is remarkably different from the other events. Because this event took place during daytime hours, under clear sky and windy conditions, TKE values were rising at a steady rate until 09:30 LST

(not shown), when $\theta_v$ started to decrease. From this moment, TKE correspondingly decreased during most of the period under DMC activity (except between 11:00 and 11:30 LST) in response to generally lighter winds (less mechanical production) and cooler surface temperatures (damping buoyancy production).

## 6   Conclusions

The time evolution of atmospheric variables and scalar fluxes during the occurrence of surface outflows produced by deep

convective storms in a tropical rainforest was analyzed utilizing high-frequency, multi-level measurements performed at the 80-m walk-up tower of the Amazon Tall Tower Observatory (ATTO) located in northern Brazil. Four convective outflows that passed over ATTO from late October to mid-November of 2015 were studied, with three of them occurring during the early evening transition or nighttime hours and one during the early afternoon. The evening/nocturnal events were characterized by well-defined gust fronts associated with moderate decreases in virtual potential temperature and increases in wind speed. In

contrast, the early afternoon event was a weak outflow, lacking a sharp gust front and producing only a slight drop in virtual potential temperature. With the gust front arrival, positive sensible heat flux ($H$) was enhanced, possibly due to sinking of colder air. This behavior was mainly observed at above-canopy levels in the three evening/nocturnal events; within the canopy the perturbations in $H$ caused by the outflow were weaker. Following the period with prevailing positive values, $H$ experienced a significant change becoming negative in the wake of the storms, characterizing an enhanced nocturnal regime. The highly

turbulent nature of the convective outflows was highlighted by TKE enhancements accompanying the passage of the gust fronts over ATTO, with TKE values during this period being comparable to those observed in grass fire experiments. As for the latent heat flux ($LE$), it increased right after the gust front in response to drier air coming from the outflow. The high-frequency, multi-level data and quantitative analyses of enhanced heat fluxes and associated intense turbulence caused by storm outflows in a rainforest presented in this study help not just to better document the complex interactions between storm-modified air

masses and forest canopy, but also highlight features that are challenging, or perhaps impossible, to measure based solely on conventional observational platforms. More specifically, the observations of highly positive $H$ flux and TKE magnitude could be used to qualitatively and quantitatively verify numerically simulated gust front interactions with the lower PBL in forested regions. To summarize our results, Figure 8 depicts a conceptual model for the time evolution of $H$ above and within the canopy for the evening/nocturnal gust front events, with t1 being representative of pre-gust front conditions and t2 representative of

the wake of the convective storms.



*Author contributions.* MIO, AOM and MOA developed the scientific idea of the study and project. OCA, MS, PESO and AT took part in data collecting and analysis and provided the scientific support on micrometeorological issues. ELN provided scientific support on severe weather concepts. MIO and DVB analyzed the data. All authors contributed to the discussion and interpretation of the results.

*Competing interests.* The authors declare that they have no conflict of interest.

5  *Acknowledgements.* This work was developed during the first author's participation in the ATTO project under Grant 574009/2008-6, process: 381948/2009-9 (Dec 2015-Jul 2017) of the Conselho Nacional de Desenvolvimento Científico e Tecnológico (CNPq). For the operation of the ATTO site, we acknowledge the support by the German Federal Ministry of Education and Research (BMBF contract 01LB1001A) and the Brazilian Ministério da Ciência, Tecnologia e Inovação (MCTI/FINEP contract 01.11.01248.00) as well as the Amazon State University (UEA), FAPEAM, LBA/INPA and SDS/CEUC/RDS-Uatumã. Alessandro Araújo, Marta Sá and the Micrometeorology group from
10 LBA project in Manaus have been responsible for collecting, organizing and quality controlling part of the data used in the study. This work was in particular supported by the Max Planck Society (MPG), the Instituto Nacional de Pesquisas da Amazônia (INPA), CNPq and CAPES.



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





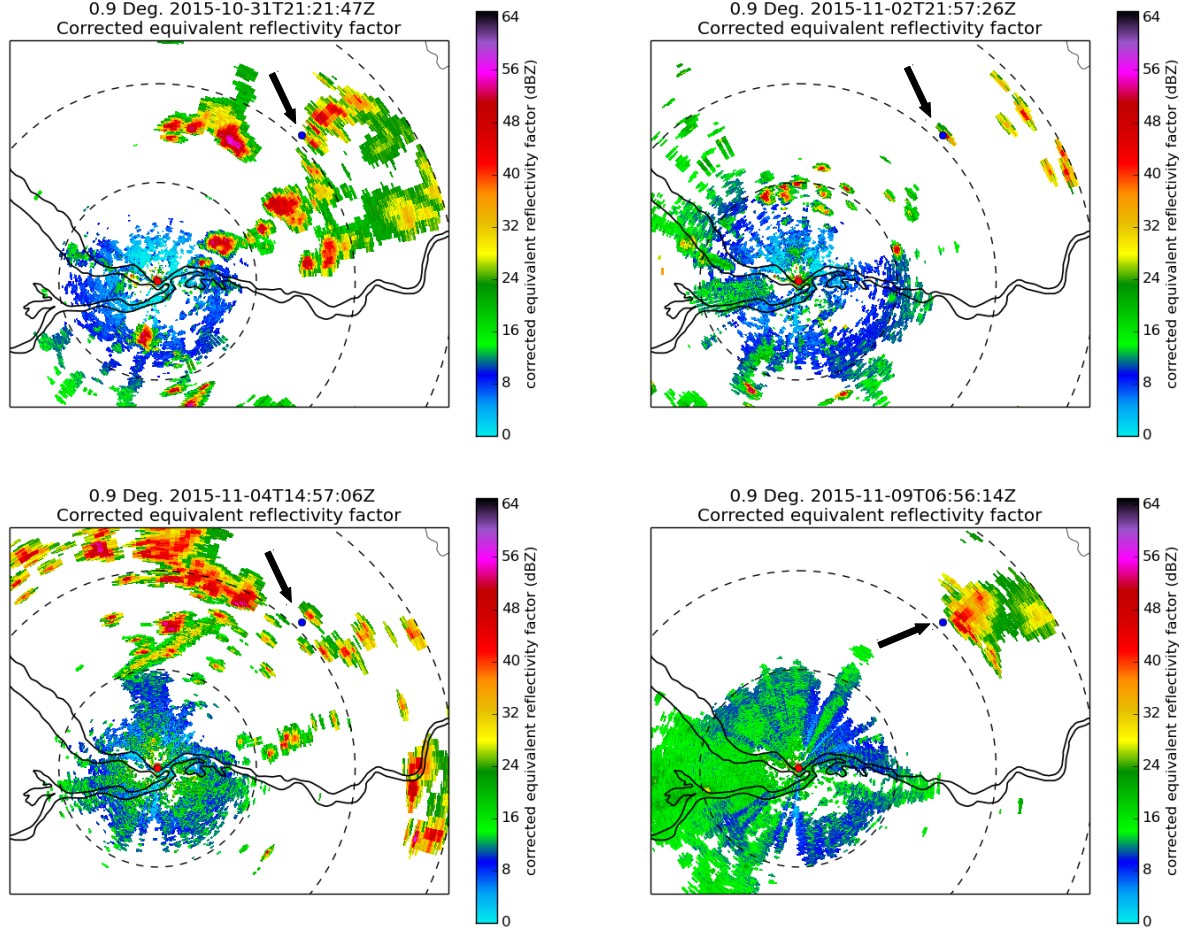

**Figure 1.** 0.9° PPI reflectivity imagery from the Manaus Doppler radar for the four DMC events studied. (upper left) 1721 on LST 31 Oct 2015, (upper right) 1757 LST on 2 Nov 2015, (lower left) 1057 LST on 4 Nov 2015, and (lower right) 0256 LST on 9 Nov 2015. The red (blue) dot shows the location of Manaus (ATTO tower). The black arrow indicates the convective elements that were sampled at the walk-up tower.







**Figure 2.** Temporal evolution of the mean horizontal wind speed at the different vertical levels, according to legend, for event 1 (a), event 2 (b), event 3 (c) and event 4 (d). The dashed vertical lines indicate the passage of the storm over or near the site.





**Figure 3.** The same as in Fig. 2, but for virtual potential temperature.



**Figure 4.** The same as in Fig. 2, but for sensible heat flux.





**Figure 5.** The same as in Fig. 2, but for air and leaf surface temperature.






**Figure 6.** The same as in Fig. 2, but for latent heat flux.





**Figure 7.** The same as in Fig. 2, but for turbulent kinetic energy.





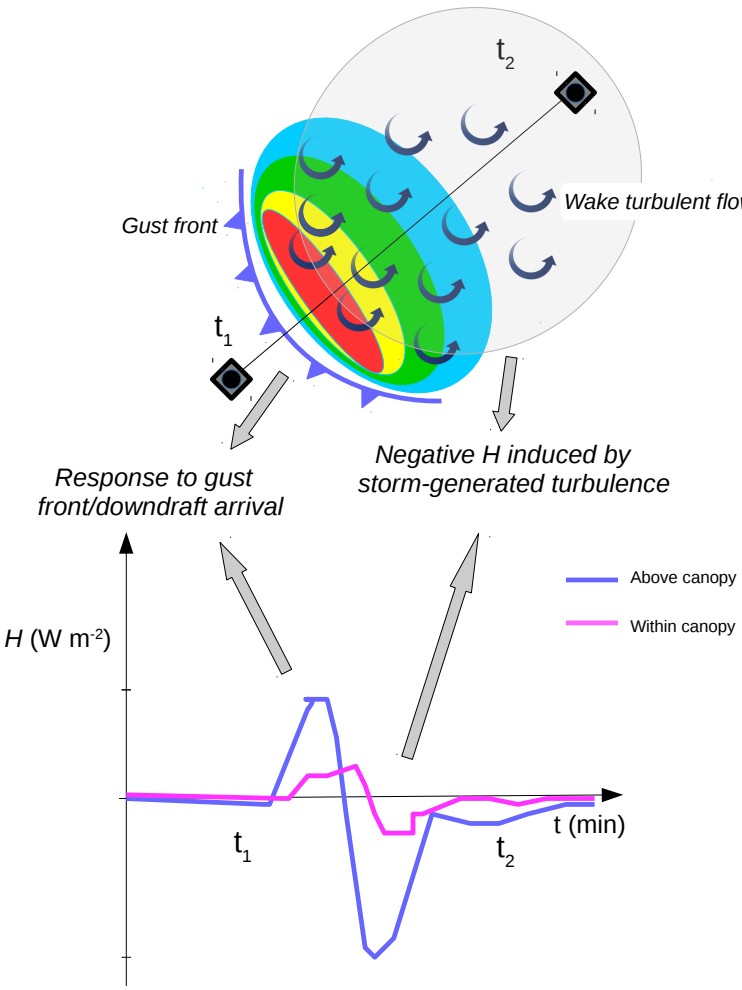

**Figure 8.** Schematics illustrating the effects of a gust front passage over a tall tower in the forest during nighttime hours. Top: a gust front (blue line with triangles) from a convective storm (color-shaded ellipsoids; cold [warm] colors represent low [high] radar reflectivity values) approaches the tall tower (gray square) at t1. At t2, the gust front has passed by the tower site which now is embedded in the cold pool's turbulent wake (gray large ellipsoid with circular arrows representing turbulent eddies). Bottom: corresponding sensible heat flux response to gust front passage at tower levels above (blue) and below (pink) the canopy.



**Table 1.** Main characteristics of the four storm events investigated in this study. Asterisks indicate SBMN (lower resolution) operational soundings.

| Date | Event duration (LST) | Echo characteristics | Max. gust front $V_h$ (m s$^{-1}$) | Max. $\theta_v$ drop (K) | Raobs (UTC) |
|---|---|---|---|---|---|
| 31 October | 1600-0000 | Multicell cluster | 8 | 8 | 0000* |
| 2 November | 1700-0000 | Isolated cell | 10 | 6-8 | 1725 |
| 4 November | 0930-1230 | Scattered cells | 4 | 3.5 | 1329 |
| 9 November | 0100-0600 | Multicell cluster | 9 | 4 | 0000* |