# Peer review of "Planetary boundary layer evolution over the Amazon rain forest in episodes of deep moist convection at ATTO"

_Atmospheric Chemistry and Physics, 2019_

## Referee Comment (RC1) · Anonymous Referee #1 · 29 May 2019

Review: Planetary boundary layer evolution over the Amazon rain forest in episodes of deep moist convection at ATTO

The authors present a short case study regarding the passage of convective systems at the ATTO site and corresponding measurements of turbulent quantities at the ATTO site in the Brazilian Amazon. While the topic is of general interest to the community, I have strong methodological concerns regarding the turbulence measurements during convective episodes, which are not relieved by reading the methods section. It is imperative that they are addressed before publication, which may not be possible.

General / Major Comments:

1) My main comment, which needs to be addressed before publication is due to methdology. Section 4, which is the main results section investigates turbulent fluxes and TKE during the passage of storm systems. However, I am not convinced that the data during these episodes is reliable and supports the conclusions. During rain events or with water on the transducers CSAT3 do not work very well. While light rain may be acceptible, during heavy rain (>3 or so mm/h) sonic anemometers generally produce no accurate readings. There may also be an issue with virbrations of sensor mounts and tower that affects measurements during storms. For example I find the reported values of TKE (increase by factor of $\sim$50 during passage of cells) and H (up to -800 W/m2) questionable/ unrealistic. Can values like this be supported from the literature. The methodology does not mention any kind of data quality assurance. For example, the authors should look at turbulence spectra to check whether these look OK and eliminate data observed during rain events or during periods when sonic transducers are likely wet.

2) The paper presents 4 events (mostly with time series of theta, U and other variables during the course of the event), but it is not clear to what extent atmospheric behavior during these events is generalization. Are these events the norm, or are they unusual. I feel that this severely limits the knowledge that can be gained from this work.

Specific: P2L12: "Much of the knowledge on the effects of DMC on PBL evolution has been gained from research based on the GARP" > I suggest to modify this statement, as it sounds as if this experiment delivered a majority of knowledge on the topic.

Section 2.1: Given that the study concerns DMC, the authors should expand here on their treatment of periods with rain. Rainfall and water on CSAT3 transducers impacts turbulence measurements. How was this dealt with? Are there any longer datasets available? For example, the work described in Fuentes et al has 9 levels of turbulence between 0.5 and 55 m and data is collected for $\sim$ 1 year.

P4L3: "The study period extended from 29 October 2015 to 20 November 2015" > I

have a question regarding the study period. I know that this site is used extensively for research (mainly Atmospheric Chemistry). I am a bit surprised that there is only <1 month of data available for turbulence measurements. Could the authors elaborate on the deployment of the CSAT3s.

P5L9: "Following the aforementioned procedure, four DMC events were selected for investigation" > It would be good if the authors could provide some measure of how many systems there were in total. I understand that this work more or less presents case studies, but I feel some quantification of events should be done.

Table 1: Are there other measures that could be included, such as cloud brightness temperature/ cloud top height or precipitation to get a sense of the strength. The Table caption should indicate where Vh and theta_v where measured, as well as location of RAOBS

LP6L18: "In this situation, the establishment of a shallow, cool near-surface stable layer occurs earlier than it would be the case for a typical undisturbed diurnal cycle." > This may or may not be true, but 18 LST is roughly the time of sunset, so I am not sure to what extent this really constitutes and "early nightfall" because from this work, we don't know what the normal transition looks like.

P6L30: "As the gust front impacted the tower after sunset, an early nightfall effect was also observed, similar to event 1." > I don't understand this. I thought an early nightfall means that there is no recovery since there is no additional energy input in the system that can lead to recovery, but this Figure 3b does show that theta recovers.

P7L8: "very stable stratification" > can this be quantified. if not, I suggest to remove the "very"

P8L2: "An "attempt" of a recovery phase was observed as a slight increase in theta_v around 04:00" > I don't find this very convincing. What is different at 4:00 to lets say 5:00.

Figure 3d: Why does theta_v at 55m and 40m behave so differently, between 3:30 and 5:00. Can you make sure that this is not an issue with the data.

Section 4: I feel that there are very likely methodological issues with this section. We know that CSAT3 analyzers don't work well during (strong) rain. Also, storms might introduce vibrations to tower and sensor mounts that affect 'observed' H. In summary much care needs to be taken to make sure that the findings in this section are robust. I feel that the increase in H is consistent with the cooling of the air and a surface response. At the same time, I find sustained fluxes of -800 W/m2 for several minutes surprising (Figure 4b). Especially since before and after the passage of the front, fluxes are +/- zero. I would feel much more confident, if the authors could back up their findings with a comparison to H fluxes observed during other studies. Also if fluxes are integrated to 30 minutes (which is the conventional standard). Do they make sense? This problem affects Figures 4,6,7 as all these rely on data from the CSAT3s. One indication of issues with the data is for example, that Vh changes from ∼3-4 to 10m/s (factor of 3) during the passage from the first storm, but observed TKE goes from 0.1 (?) to 6 m2/s2, which is a factor of 60. I am don't think that this is real.

Technical: P2L10: "into the surface" > "into the ABL" or "towards the surface" P3L8: "engender the venting" > affect the venting P4L34: "BLIS" > consider writing out for readability. I had already forgotten what BLIS stood for and had to look it up. P6L17: "an effective" > this does not work very well in English (since it sounds as if the nightfall is effective" Maybe: "a situation akin to an early nightfall" ?

---

## Referee Comment (RC2) · Anonymous Referee #2 · 29 Aug 2019

**Comments on "Planetary boundary layer evolution over the Amazon rain forest in episodes of deep moist convection at ATTO", by Oliveira et al.**

August 29, 2019

**General remarks**

This manuscript analyzes turbulence data measured at several levels at an 80-m high tower at the ATTO site in the Amazon. The data are analyzed around the occurence of deep moisture convection (DMC) and strong downdrafts of cold air from above associated with the passage of storms by the tower, as identified by meteorological radar.

The manuscript is well written and easy to understand. It is also basically an observational study. It consists of the description of the evolution in time of the response in atmospheric variables measured by the tower to the passage of the pool of cold air from the storm downdrafts.

As such, the manuscript does not contain new quantitative theories, nor does it describe any new phenomena, with the possible exception of the detailed attention to the temporal behavior of the sensible and latent heat fluxes, and the turbulence kinetic energy, measured at several levels at the tower during those episodes. It is however useful as a good example of the application of high-quality research data to the understanding of influence of storm downdrafts on the planetary boundary layer. As such, I think it deserves publication.

Because it is well written and documented, and of its descriptive nature, there are very few remarks that I can make on the manuscript. They are listed in the specific comments below.

**Recommendations**

Publish.

**Specific comments**

p. 2, l. 19–20    "This stage initiates in the wake of the storms and **it** may take 7-10 h for the ML to re-establish undisturbed conditions."

p.2, l. 32–33    "They found latent and sensible heat flux enhancements reaching peak values of $60\,\mathrm{W\,m^{-2}}$ and $250\,\mathrm{W\,m^{-2}}$ for large, organized Mesoscale Convective Systems (MCSs)."

In general $LE \gg H$ over the ocean, but here you are saying $\Delta LE = 60\,\mathrm{W\,m^{-2}}$, $\Delta H = 250\,\mathrm{W\,m^{-2}}$. Please confirm that the enhancements are much larger for $H$.

p.4, l 10–11    "Computation of turbulent quantities from tower data such as mean flow, heat fluxes and turbulent kinetic energy were accomplished by employing Reynolds averaging at 1-min time intervals."

Strictly speaking, "Reynolds averaging" is ensemble averaging, for which the so-called Reynolds postulates apply. Here, you should say "time averages over 1-min. intervals".

p.4, l 31    gust (not gusts) fronts.

---

## Referee Comment (RC3) · Kathleen Schiro (Referee) · 6 Sep 2019

Review of "Planetary boundary layer evolution over the Amazon rain forest in episodes of deep moist convection at ATTO" by M. Oliveira et al.

This study uses data from a tall tower in the Amazon to assess the thermodynamic and kinematic properties of convective downdrafts/outflows/cold pools. The study focuses on four deep convective cases of differing spatial characteristics. Three of the four cases were nocturnal, while one occurred during the early afternoon hours. The authors find interesting differences between the thermodynamic and kinematic properties of the PBL after the different convective system passages. Notable differences include (1) well-defined gust fronts in the nocturnal cases vs. a weakly defined gust front in the daytime case; (2) different PBL layers recover quite differently after system passage for the isolated system cases; (3) nighttime cases have clearly defined increases in sensible heat near the time of gust front arrival and decreases afterwards, whereas the daytime case exhibits different behavior. Interesting differences are noted in the response of the surface layer of the PBL vs. the top of the canopy, including that heat fluxes are most pronounced above the canopy rather than within the canopy.

I think this study is well-written and presents many interesting findings. The authors provide insightful discussions throughout. The authors' findings are complementary to past studies, yet provide new insights into processes that are difficult to observe and are thus not readily studied (downdrafts, PBL dynamics and thermodynamics, detailed land-atmosphere interactions).

Overall, I recommend that this study be published in ACP with **_minor revisions_**.

General comments:

1. You provide various explanations for defining and choosing your cases. You also attempt to explain why you chose such a short study period on page 4. However, your explanations seem rather unclear to me. More specifically, could you clarify what you mean by "We have chosen such a short time window primarily because of the nonstationary nature of the events under study, but also to avoid contamination from low-frequency, non- turbulent processes, and, therefore, guarantee that the discussion refers to turbulent quantities alone (lines 11-14, page 4)"? Stating that "Only storms that produced detectable impacts on the evolution of meteorological variables at the tower site were selected (p. 4, lined 28-29)" makes sense over such a short time period, but again, I don't feel that the short time period is ever adequately justified.
2. Since it's hard to generalize day vs. night, organized vs. disorganized convection differences in PBL behavior following system passage when you only have four cases, I think you should add a few concluding sentences cautioning the readers against generalizing these conclusions. Perhaps an appropriate place to do so is after the schematic is introduced in the conclusion?

Specific comments:

Lines 9-10: Please revise to read "The nocturnal events had well-defined gust fronts with moderate decreases in virtual potential temperature and increases in wind speed."

Line 12: "experienced an increase" – how about just "increased"?

Page 5, line 21: Schiro and Neelin (2018, ACP) compare statistics on downdraft/cold pool properties from both sub-MCS size system and MCS systems at the GoAmazon2014/5 site. Wang et al. (2019) also uses GoAmazon2014/5 data to look at cold pool/downdraft characteristics. Both studies use the S-Band radar to classify the deep convection. It seems that references to these studies could be appropriate here.

Figure 1: It would be very helpful to add spatial information to the axes on the subpanels, especially since you discuss the degree of spatial organization. Also, please mention what the circles (dashed lines) mean in the caption (what distance is this from the radar?). Lastly, please label the panels a-d.

Oct 31 case – It seems to me (from Fig. 1) that this exhibits a decent amount of organizational structure (leading line, trailing stratiform), even though the individual leading-edge cells passing over the tower may have seemed disorganized or separated from one another at any given time or may have merged with other isolated cells (as you mention). The thermodynamic and dynamic responses (Figs. 2 and 3) also suggest that this is an MCS. If you agree with this assessment, you may wish to revise your classification in the table and in lines 24-25 in Section 3 (p 5): "In comparison to SR98, the storms on 31 October (event 1), 2 November (event 2), and 4 November (event 3) mostly 25 resembled the unorganized arrangement that they referred to as sub-MCS-scale nonlinear systems."

P. 6, lines 9-10: You could probably reword this sentence to make it reference Figs. 2a and 3a respective to the order in which they are mentioned. Same for lines 28-29. (and pg. 7 line 26).

Page 6, line 12: What is the time of the first drop, shown in the dashed vertical line on Fig. 2a?

Page 6, line 10: I wouldn't say that the temperature decrease was significantly damped in Fig. 3a, especially if you look out past the $2^{nd}$ drop in temperature. In fact, it's interesting that the 14m temperatures seem to be lowest, whereas at 22m, they are highest (after 18:00 LST). You could maybe discuss that here and speculate why you think that might be.

P. 6, lines 29-30 – That increase in moisture is interesting. Maybe you could speculate here about why that might have occurred. Maybe it was moisture convergence occurring along the gust front edge? Saturated convective downdrafts from low levels entering a previously unsaturated PBL?

Nov 2 and Nov 8 event recovery vs. Oct 31 and Nov 9 recovery: The fact that the smaller, more isolated convective cells have a detectable PBL recovery time period than the larger MCSs, regardless of the time of day, is consistent with what we found in Schiro and Neelin (2018).

Pg. 7, line 13 – I wouldn't classify this as a drop; it's more like a "decrease," since it's rather gradual

Pg. 7 line 16: instead of "slow", how about "gradual"?

Insightful discussion in lines 16-22 of pg. 7. I agree with your assessment, since radar reflectivity at 14:57Z does seem to suggest that the cell did not pass directly over the tower.

Pg. 7, Line 24: I'd be careful about using phrases like "the most organized." It's hard to distinguish organization in the first place (though it's often loosely defined using spatial characteristics). I think classifying it as "organized" is speculative as it is, since you mention that the spatial scale is somewhere in between "isolated" and MCS. Instead, maybe you could classify it as the "system with the largest convective core"?

Fig. 3d – Why do you think the 40 m spikes are so much larger (and the data generally noisier) than at 14 and 55 m? Also, where is the rest of the data? Does missing data suggest data quality issues for this sample?

Heat flux measurements and discussion: I can't comment too much on the reliability of these data, but I don't doubt that there are noteworthy data concerns here (especially given the really large magnitudes observed in certain instances). At the very least, I think a discussion of the strengths and limitations of using these data during pre-storm and precipitating conditions is warranted in these sections.

Please explicitly define TKE and how it is computed.

References:

Wang, D., S. E. Giangrande, K. A. Schiro, M. P. Jensen, and R. A. Houze, 2019: The Characteristics of Tropical and Midlatitude Mesoscale Convective Systems as Revealed by Radar Wind Profilers. *Journal of Geophysical Research: Atmospheres,* 124(8), 4601-4619.

Schiro, K. A. and J. D. Neelin, 2018: Tropical Continental Downdraft Characteristics: Mesoscale Systems versus Unorganized Convection. Atmospheric Chemistry and Physics, 18, 1997-2010.

-Kathleen Schiro

---

## Author Comment (AC1) · 17 Oct 2019

**Reply to Reviewer 1**

**Major comments**

**Reviewer says:** *1) My main comment, which needs to be addressed before publication is due to methodology. Section 4, which is the main results section investigates turbulent fluxes and TKE during the passage of storm systems. However, I am not convinced that the data during these episodes is reliable and supports the conclusions. During rain events or with water on the transducers CSAT3 do not work very well. While light rain may be acceptable, during heavy rain (>3 or so mm/h) sonic anemometers generally produce no accurate readings. There may also be an issue with vibrations of sensor mounts and tower that affects measurements during storms. For example I find the reported values of TKE (increase by factor of ~50 during passage of cells) and H (up to -800 W/m2) questionable/ unrealistic. Can values like this be supported from the literature. The methodology does not mention any kind of data quality assurance. For example, the authors should look at turbulence spectra to check whether these look OK and eliminate data observed during rain events or during periods when sonic transducers are likely wet.*

**Reply**: This is a valid concern. This issue needs to be addressed to give the readers confidence in the results. We are confident in them, and these are the main reasons:

i.        Precipitation was never large. Total precipitation along the entire duration of the events was 2.3, 1.0, 5.3 and 1.5 for events 1 to 4 respectively, therefore only in event 3 exceeding the limit mentioned by the reviewer for "heavy rain". We are now including this fact in the text and adding the plot below, showing precipitation evolution along each event, as a supplementary figure.

ii.        Nevertheless, it did rain in all cases and there is also the issue raised by the reviewer regarding vibration of the mounts and tower. To address that, and following the suggestion from the reviewer, we plotted TKE spectra and heat flux cospectra for the 4 different portions of events 1 and 2: before the gust front (I); the period of upward heat flux that marks the gust front arrival (II); the period of large downward heat flux that corresponds to enhanced storm-generated turbulence (III) and the wake period after the event (IV). This is only done for events 1 and 2, because these are the cases when these periods can be easily identified. The plots are shown below (Figs. $R_1 2$ to $R_1 5$). It is clear that the TKE spectra and heat flux cospectra are, in all cases, well-organized, tending to zero in the high-frequency limit, indicating that there is reduced levels of noise. Besides, the upward or downward fluxes happen over the entire range of turbulence scales, being well organized vertically as well. It gives us a high degree of confidence in our dataset. These plots have also been included as supplementary material and a discussion referring to them has been included in the main text. In Figs. $R_1 6$ and $R_1 7$, the raw velocity and temperature turbulent data from events 1 and 2 are also shown, indicating the absence of spikes and random fluctuations. They have also been included as supplementary material. Paragraphs explaining that the data quality analysis is shown in the supplementary material have also been added to the main manuscript.

[Figure]

Fig. R$_1$1. 1-minute and total precipitation for each event.

[Figure]

Fig. R$_1$2. Multiresolution TKE spectra for the 4 periods of event 1.

[Figure]

Fig. R₁3. The same as in Fig. R₁2, but for heat flux cospectra.

[Figure]

Fig. R₁4. The same as in Fig. R₁2, but for event 2.

[Figure]

Fig. R$_1$5. The same as in Fig. R$_1$3, but for event 2.

[Figure]

Fig. R$_1$6. Time series of the velocity components and temperature along event 1.

[Figure]

Fig. $R_1$7. Time series of the velocity components and temperature along event 2.

iii.    Regarding the large observed values of both TKE and heat flux, it is important to stress that these values refer to transient events, and they have been determined using 1-min time windows. The spectra and cospectra shown in Figs. $R_1$2-5 show that this time window captures the majority of the turbulent fluctuations. Transient events such as these may, indeed, have very large magnitudes, and still be genuine. Certainly, the average flux determined over a more typical 30-min window in would have a much smaller magnitude in these cases, but it would miss all the dynamics of the event passage. There are previous observations from the literature that support these values. Hohenegger and Bretherton (2011) reported observed values of PBL-averaged TKE during cases of deep convection in ARM and KWAJEX experiments. TKE values that exceed 10 $m^2/s^2$ are common during cases of deep convection. We have not found published observations of vertical sensible heat fluxes as large as those we are reporting, but this is precisely one of the main objectives of the paper: to report this type of observations for the first time. However, in a previous study of our group, we reported similarly high transient fluxes of sensible heat in the horizontal direction, this time caused by the advance of an air mass with distinct characteristics along the surface of a river (Acevedo et al., 2007). Besides, we also have observations taken during GO-Amazon project that show heat flux evolutions and magnitudes that are similar to those being presently reported (Fig. 6.2 in Oliveira (2017), in Portuguese). This is the Doctorate thesis of one of the coauthors, Pablo Oliveira, where these GO-Amazon events have also been simulated using a simple column model that uses K-theory to predict the fluxes, indicating that the very large thermal gradients and wind speeds observed during the transient events may indeed drive very large fluxes, although for a very brief period. TKE is also very large in these observations, reaching 12 $m^2/s^2$.

**Reviewer says:** *2) The paper presents 4 events (mostly with time series of theta, U and other variables during the course of the event), but it is not clear to what extent atmospheric behavior during these events is generalization. Are these events the norm, or are they unusual. I feel that this severely limits the knowledge that can be gained from this work.*

**Reply**: Yes, the reviewer is correct. We have added the following sentences at the end of the conclusion to make it clear that we are not claiming that the results are general:

> Despite the consistency found among the events analyzed, it is important to stress that the study is based on a reduced number of events (4) and that a more detailed analysis with a larger number of cases is necessary to validate the conclusions. They will be possible along ATTO project, when continuous turbulence observations will be available from the surface to 320 m.

**Reviewer says:** *Specific: P2L12: "Much of the knowledge on the effects of DMC on PBL evolution has been gained from research based on the GARP" > I suggest to modify this statement, as it sounds as if this experiment delivered a majority of knowledge on the topic.*

**Reply**: Yes, the reviewer is correct, although we believe GATE was extremely relevant in the early developments on the field. We reworded it to "*Much of the initial knowledge on the...*"

**Reviewer says:** *Section 2.1: Given that the study concerns DMC, the authors should expand here on their treatment of periods with rain. Rainfall and water on CSAT3 transducers impacts turbulence measurements. How was this dealt with? Are there any longer datasets available? For example, the work described in Fuentes et al has 9 levels of turbulence between 0.5 and 55 m and data is collected for _ 1 year.*

**Reply**: The issue regarding rainfall has been addressed in the reply to major comment (1), above. The dataset used in this paper comes from an Intensive Operating Period (IOP) at the ATTO site. This was carried out before most of the instruments were deployed for the continuous measurements (scheduled to happen in the upcoming months). Although data from the GO-Amazon project could be used for comparison, it has not been done. It presently focuses on case studies, and for this purpose the ATTO IOP dataset has the advantage of a deeper vertical coverage as compared to GO-Amazon. Such a comparison is certainly a good idea for future work.

**Reviewer says:** *P4L3: "The study period extended from 29 October 2015 to 20 November 2015" I have a question regarding the study period. I know that this site is used extensively for research (mainly Atmospheric Chemistry). I am a bit surprised that there is only 1 month of data available for turbulence measurements. Could the authors elaborate on the deployment of the CSAT3s.*

**Reply**: As mentioned previously, the dataset correspond to an IOP carried out in 2015. As of October 2019, the full micrometeorological instrumentation have not yet been deployed, and the continuous observations are scheduled to start early in 2020. Although some levels have operated continually for a long time, it is only during this IOP that there has been multiple CSATs operating simultaneously. In reply to comment 2, above, we have added a sentence stating the relevance of the upcoming continuous measurements for the generalization of the present results. It has also been added to the manuscript that the period of observations corresponded to an IOP.

**Reviewer says:** *P5L9: "Following the aforementioned procedure, four DMC events were selected for investigation" It would be good if the authors could provide some measure of how many systems there were in total. I understand that this work more or less presents case studies, but I feel some quantification of events should be done.*

**Reply**: The 4 cases described are the only occurrences found during the IOP. As described in the manuscript, "*Only storms that produced detectable impacts on the evolution of meteorological variables at the tower site were selected.*"

**Reviewer says:** *Table 1: Are there other measures that could be included, such as cloud brightness temperature/ cloud top height or precipitation to get a sense of the strength. The Table caption should indicate where Vh and theta_v where measured, as well as location of RAOBS.*

**Reply**: Total precipitation for each event has been included to the table.

**Reviewer says:** *LP6L18: "In this situation, the establishment of a shallow, cool near-surface stable layer occurs earlier than it would be the case for a typical undisturbed diurnal cycle." > This may or may not be true, but 18 LST is roughly the time of sunset, so I am not sure to what extent this really constitutes and "early nightfall" because from this work, we don't know what the normal transition looks like.*

**Reply**: The reviewer is correct for the cases shown when the event happens near 1800 LST, but the idea is still valid for earlier events. For that reason, we reworded the sentence to "*In this situation, the establishment of a shallow, cool near-surface stable layer may occur earlier than it would be the case for a typical undisturbed diurnal cycle.*"

**Reviewer says:** *P6L30: "As the gust front impacted the tower after sunset, an early nightfall effect was also observed, similar to event 1." I don't understand this. I thought an early nightfall means that there is no recovery since there is no additional energy input in the system that can lead to recovery, but this Figure 3b does show that theta recovers.*

**Reply**: It is a valid point. The sentence has been removed.

**Reviewer says:** *P7L8: "very stable stratification" > can this be quantified. if not, I suggest to remove the "very"*

**Reply**: "Very" has been removed from the sentence.

**Reviewer says:** *P8L2: "An "attempt" of a recovery phase was observed as a slight increase in theta_v around 04:00" > I don't find this very convincing. What is different at 4:00 to lets say 5:00.*

**Reply**: It is not much different, but the first "attempt", at 04:00 was longer and had a larger change in $\theta_v$, being therefore mentioned.

**Reviewer says:** *Figure 3d: Why does theta_v at 55m and 40m behave so differently, between 3:30 and 5:00. Can you make sure that this is not an issue with the data.*

**Reply**: The data at 40 m were, indeed, faulty. This line has been removed from the plot.

**Reviewer says:** *Section 4: I feel that there are very likely methodological issues with this section. We know that CSAT3 analyzers don't work well during (strong) rain. Also, storms might introduce vibrations to tower and sensor mounts that affect 'observed' H. In summary much care needs to be taken to make sure that the findings in this section are robust. I feel that the increase in H is consistent with the cooling of the air and a surface response. At the same time, I find sustained fluxes of -800 W/m2 for several minutes surprising (Figure 4b). Especially since before and after the passage of the front, fluxes are +/- zero. I would feel much more confident, if the authors could back up their findings with a comparison to H fluxes observed during other studies. Also if fluxes are integrated to 30 minutes (which is the conventional standard). Do they make sense? This problem affects Figures 4,6,7 as all these rely on data from the CSAT3s. One indication of issues with the data is for example, that Vh changes from ~3-4 to 10m/s (factor of 3) during the passage from the first storm, but observed TKE goes from 0.1 (?) to 6 m2/s2, which is a factor of 60. I am don't think that this is real.*

Reply: This issue has been addressed in the reply to major comment 1, above.

**Reviewer says:** *Technical: P2L10: "into the surface" > "into the ABL" or "towards the surface" P3L8: "engender the venting" > affect the venting P4L34: "BLIS" > consider writing out for readability. I had already forgotten what BLIS stood for and had to look it up. P6L17: "an effective" > this does not work very well in English (since it sounds as if the nightfall is effective" Maybe: "a situation akin to an early nightfall" ?*

Reply: Done.

**REFERENCES:**

Acevedo, O.C., O.L. Moraes, R. da Silva, V. Anabor, D.P. Bittencourt, H.R. Zimmermann, R.O. Magnago, and G.A. Degrazia, 2007: Surface-to-Atmosphere Exchange in a River Valley Environment. J. Appl. Meteor. Climatol., 46, 1169–1181, https://doi.org/10.1175/JAM2517.1

Hohenegger, C. and Bretherton, C. S.: Simulating deep convection with a shallow convection scheme, Atmos. Chem. Phys., 11, 10389-10406, https://doi.org/10.5194/acp-11-10389-2011, 2011.

Oliveira, P. E. S. (2017) Estudo da turbulência atmosférica na floresta Amazônica - análise de dados micrometeorológicos e modelagem numérica (Doctoral dissertation). Available at http://repositorio.ufsm.br/handle/1/14596

---

## Author Comment (AC2) · 17 Oct 2019

Title: Planetary boundary layer evolution over the Amazon rain forest in episodes of deep moist convection at ATTO.

Manuscript Number: acp-2019-373

Authors: Maurício I. Oliveira, Otávio C. Acevedo, Matthias Sörgel, Ernani L. Nascimento, Antonio O. Manzi, Pablo E. S. Oliveira, Daiane V. Brondani, Anywhere Tsokankunku, and Meinrat O. Andreae

Manuscript type: Article

Recommendation from the reviewer: Publish

**Reply to Reviewer #2:**

**General remarks**

**Reviewer says:** *This manuscript analyzes turbulence data measured at several levels at an 80-m high tower at the ATTO site in the Amazon. The data are analyzed around the occurence of deep moisture convection (DMC) and strong downdrafts of cold air from above associated with the passage of storms by the tower, as identified by meteorological radar.*
*The manuscript is well written and easy to understand. It is also basically an observational study. It consists of the description of the evolution in time of the response in atmospheric variables measured by the tower to the passage of the pool of cold air from the storm downdrafts.*

*As such, the manuscript does not contain new quantitative theories, nor does it describe any new phenomena, with the possible exception of the detailed attention to the temporal behavior of the sensible and latent heat fluxes, and the turbulence kinetic energy, measured at several levels at the tower during those episodes. It is however useful as a good example of the application of high-quality research data to the understanding of influence of storm downdrafts on the planetary boundary layer. As such, I think it deserves publication.*

*Because it is well written and documented, and of its descriptive nature, there are very few remarks that I can make on the manuscript. They are listed in the specific comments below.*

**Reply:** The authors would like to thank the reviewer for positive remarks.

**Specific comments**

**Reviewer says:** *p. 2, l. 19–20 "This stage initiates in the wake of the storms and **it** may take 7-10 h for the ML to re-establish undisturbed conditions."*

**Reply:** The sentence has been changed.

**Reviewer says:** *p.2, l. 32–33 "They found latent and sensible heat flux enhancements reaching peak values of 60 W m$^{-2}$ and 250 W m$^{-2}$ for large, organized Mesoscale Convective Systems (MCSs)."*

*In general LE $\gg$ H over the ocean, but here you are saying $\Delta LE$ =60 Wm$^{-2}$ , $\Delta H$ = 250 Wm$^{-2}$ . Please confirm that the enhancements are much larger for H.*

**Reply:** The reviewer is correct. We rephrase the sentence to *"They found sensible and latent heat flux enhancements reaching peak values of 60 W m$^{-2}$ and 250 W m$^{-2}$ for large, organized Mesoscale Convective Systems (MCSs)."*

**Reviewer says:** *p.4, l 10–11 "Computation of turbulent quantities from tower data such as mean flow, heat fluxes and turbulent kinetic energy were accomplished by employing Reynolds averaging at 1-min time intervals."*

*Strictly speaking, "Reynolds averaging" is ensemble averaging, for which the so-called Reynolds postulates apply. Here, you should say "time averages over 1-min. intervals".*

**Reply:** It is a valid point. The sentence has been changed.

**Reviewer says:** *p.4, l 31 "gust (not gusts) fronts."*

**Reply:** The sentence has been changed.

---

## Author Comment (AC3) · 17 Oct 2019

Title: Planetary boundary layer evolution over the Amazon rain forest in episodes of deep moist convection at ATTO.

Manuscript Number: acp-2019-373

Authors: Maurício I. Oliveira, Otávio C. Acevedo, Matthias Sörgel, Ernani L. Nascimento, Antonio O. Manzi, Pablo E. S. Oliveira, Daiane V. Brondani, Anywhere Tsokankunku, and Meinrat O. Andreae

Manuscript type: Article

Recommendation from the reviewer: Minor revisions

**Replies to Reviewer #3 (Dr. Kathleen Schiro):**

This study uses data from a tall tower in the Amazon to assess the thermodynamic and kinematic properties of convective downdrafts/outflows/cold pools. The study focuses on four deep convective cases of differing spatial characteristics. Three of the four cases were nocturnal, while one occurred during the early afternoon hours. The authors find interesting differences between the thermodynamic and kinematic properties of the PBL after the different convective system passages. Notable differences include (1) well-defined gust fronts in the nocturnal cases vs. A weakly defined gust front in the daytime case; (2) different PBL layers recover quite differently after system passage for the isolated system cases; (3) nighttime cases have clearly defined increases in sensible heat near the time of gust front arrival and decreases afterwards, whereas the daytime case exhibits different behavior. Interesting differences are noted in the response of the surface layer of the PBL vs. the top of the canopy, including that heat fluxes are most pronounced above the canopy rather than within the canopy.

I think this study is well-written and presents many interesting findings. The authors provide insightful discussions throughout. The authors' findings are complementary to past studies, yet provide new insights into processes that are difficult to observe and are thus not readily studied (downdrafts, PBL dynamics and thermodynamics, detailed land-atmosphere interactions).

Overall, I recommend that this study be published in ACP with minor revisions.

**The authors deeply appreciate the in-depth critics and suggestions provided by the reviewer. We believe the manuscript has been significantly improved as a result of this revision. Below the reviewer will find our point-by-point responses, written in bold-faced dark blue.**

**General comments:**

1. You provide various explanations for defining and choosing your cases. You also attempt to explain why you chose such a short study period on page 4. However, your explanations seem rather unclear to me. More specifically, could you clarify what you mean by "We have chosen such a short time window primarily because of the nonstationary nature of the events under study, but also to avoid contamination from low- frequency, non- turbulent processes,

and, therefore, guarantee that the discussion refers to turbulent quantities alone (lines 11-14, page 4)"? Stating that "Only storms that produced detectable impacts on the evolution of meteorological variables at the tower site were selected (p. 4, lined 28-29)" makes sense over such a short time period, but again, I don't feel that the short time period is ever adequately justified.

**We agree with the reviewer that both the choice of the period of study as well as the use of short averaging windows can be further explained and clarified. These points are addressed below.**

**Period of study: The dataset used in this paper refers to an Intensive Operating Period (IOP) at the ATTO site focused on the period from late October through mid-November 2015. At the time this IOP was conducted, most of the instruments had not been were deployed for continuous measurements; this is scheduled to happen in the upcoming months. Nevertheless, only during this IOP, there was multiple micrometeorological instruments (CSATs) operating simultaneously at several tower levels, making this period suitable for conducting the case studies we presented. We have added to the manuscript that the period of observations refer to an IOP.**

**Averaging time window: The short, 1-min time window we describe in lines 11-14 (pg. 4) refers to the averaging time interval from which turbulent fluctuations are calculated from. Such short averaging time window is needed to capture the dynamics of the gust front passage given the highly transient, abrupt nature of the phenomenon. Average flux calculations determined over a more typical 30-min window would yield much smaller flux magnitude in the cases studied, i.e., introducing the adverse effect of smoothing out the flux peaks and thus, missing all the dynamics of the event passage.**

2. Since it's hard to generalize day vs. night, organized vs. disorganized convection differences in PBL behavior following system passage when you only have four cases, I think you should add a few concluding sentences cautioning the readers against generalizing these conclusions. Perhaps an appropriate place to do so is after the schematic is introduced in the conclusion?

**Thank you for the comment. This concern, also raised by Reviewer #1, is a relevant suggestion which helps to present our conclusions more clearly and caution readers about the generarity of our findings. Motivated by your suggestion, we have included the following statements in the conclusion:**

**"*Despite the consistency found among the events analyzed, it is important to stress that the study is based on a reduced number of events (4) and that a more detailed analysis with a larger number of cases is necessary to validate the conclusions. They will be possible along ATTO project, when continuous turbulence observations will be available from the surface to 320 m.*"**

**Specific comments:**

Lines 9-10: Please revise to read "The nocturnal events had well-defined gust fronts with moderate decreases in virtual potential temperature and increases in wind speed."

**The sentence has been modified as suggested.**

Line 12: "experienced an increase" – how about just "increased" ?

**The modification has been done.**

Page 5, line 21: Schiro and Neelin (2018, ACP) compare statistics on downdraft/cold pool properties from both sub-MCS size system and MCS systems at the GoAmazon2014/5 site. Wang et al. (2019) also uses GoAmazon2014/5 data to look at cold pool/downdraft characteristics. Both studies use the S-Band radar to classify the deep convection. It seems that references to these studies could be appropriate here.

**Thank you for pointing that out. Your comment has motivated us to rephrase a couple of sentences in the manuscript. On page 4 we have added a citation to Schiro and Neelin (2018) when mentioning previous studies that have applied quantitative criteria to select the convective events. On page 5 we now cite both Schiro and Neelin (2018) and Wang et al. (2019) together with SR98.**

Figure 1: It would be very helpful to add spatial information to the axes on the subpanels, especially since you discuss the degree of spatial organization. Also, please mention what the circles (dashed lines) mean in the caption (what distance is this from the radar?). Lastly, please label the panels a-d.

**We agree that relevant spatial information was lacking in the subpanels and caption of Figure 1; in the new version such information is provided. Thank you.**

Oct 31 case – It seems to me (from Fig. 1) that this exhibits a decent amount of organizational structure (leading line, trailing stratiform), even though the individual leading-edge cells passing over the tower may have seemed disorganized or separated from one another at any given time or may have merged with other isolated cells (as you mention). The thermodynamic and dynamic responses (Figs. 2 and 3) also suggest that this is an MCS. If you agree with this assessment, you may wish to revise your classification in the table and in lines 24-25 in Section 3 (p 5): "In comparison to SR98, the storms on 31 October (event 1), 2 November (event 2), and 4 November (event 3) mostly 25 resembled the unorganized arrangement that they referred to as sub-MCS-scale nonlinear systems."

**Thank you very much for raising this important point, but during this event we found no contiguous region of reflectivity above 30 dBZ displaying 100 km or more in length. To further verify if an MCS could be characterized in any given moment of the evolution of this event, we checked the GOES-13 thermal IR imagery during the life**

cycle of the storm system, but the only MCS observed in that period was located in northern Pará state, hundreds of km to the northeast of the region of interest. To illustrate that, we are copying, in this reply, the GOES-13 enhanced thermal IR image valid around the time of the radar image shown in Fig. 1a. Given these points we have no solid argument to support a claim that the event was indeed an MCS.

[Figure]

**Figure R3.1: enhanced thermal infrared GOES 13 image at 22:00 UTC 31 Oct 2015 over the Amazon region. Brightness temperatures indicated by the color shading, in °C). The yellow rectangle indicates the convective system of interest.**

P. 6, lines 9-10: You could probably reword this sentence to make it reference Figs. 2a and 3a respective to the order in which they are mentioned. Same for lines 28-29. (and pg. 7 line 26).

**We agree with your suggestion. The sentences in lines 9-10, 28-29 and 26 (pg. 7) have been reworded to properly reference Figures 2a and 3a.**

Page 6, line 12: What is the time of the first drop, shown in the dashed vertical line on Fig. 2a?

**The time of the drop represented by the dashed vertical line on Fig. 2a is 17:15 h Local Standard Time (UTC = LST + 4 h). In view of this comment, we also included the times of the drops in the caption of Figure 2 for all events. These correspond to: 17:58 LST on 2 Nov 2015 (Fig. 2b), 10:00 LST on 4 Nov 2015, (Fig. 2c) and 03:00 LST 9 Nov 2015 (Fig. 2d).**

Page 6, line 10: I wouldn't say that the temperature decrease was significantly damped in Fig. 3a, especially if you look out past the 2[nd] drop in temperature. In fact, it's interesting that the 14m temperatures seem to be lowest, whereas at 22m, they are highest (after 18:00 LST). You could maybe discuss that here and speculate why you think that might be.

**The reviewer is right when we look out after the 2[nd] drop in temperature. However, this is addressed later in the same paragraph. When we said that the temperature was damped inside the canopy, we were referring to the 1[st] drop, during the period right after the outflow starts (period II in Fig. 3a), as the drop rate of temperature at 14 and 22 m was smaller than above the canopy. In fact, temperature at 14 m was smaller than above the forest before the outflow starts, and became larger during period II.**

**The fact that temperature at 22 m is larger than at the lower levels inside the canopy is very interesting, but it is not surprising. Previous studies have shown that the temperature within the forest is consistently smaller close to the ground, especially during daytime (Viswanadham et al., 1990; Kruijt et al., 2000). This occurs because the radiative heating inside the forest starts from the canopy top towards the ground. During the night, however, we think that the energy loss at 22 m is not enough to reduce the temperature to levels below those observed close to the ground.**

P. 6, lines 29-30 – That increase in moisture is interesting. Maybe you could speculate here about why that might have occurred. Maybe it was moisture convergence occurring along the gust front edge? Saturated convective downdrafts from low levels entering a previously unsaturated PBL?

**Thank you very much for drawing our attention to these ideas. This is indeed an interesting aspect of this particular event. We agree with the reviewer's suggestions for the possible physical processes operating and, hence, we have added a new sentence taking into account these plausible hypotheses (following the "not shown" statement):**

**"*This transient moisture increase may have been caused by moisture convergence along the gust front or the intrusion of low-level saturated convective downdrafts into a previously unsaturated PBL.*"**

Nov 2 and Nov 8 event recovery vs. Oct 31 and Nov 9 recovery: The fact that the smaller, more isolated convective cells have a detectable PBL recovery time period than the larger MCSs, regardless of the time of day, is consistent with what we found in Schiro and Neelin (2018).

**Thank you again for point this out. The results regarding PBL recovery time as a function of convective mode/organization discussed in Schiro and Neelin (2018) are definitely in line with the results we found. Therefore, we have included a new paragraph at the end of subsection 3.4 and referenced Schiro and Neelin (2018) in order to shed light on the relationship between PBL recovery and convective system spatial scale.**

*"The longer recovery period observed in event 4, as well as that found in event 1, are in contrast with the short recovery observed in event 2, which points to the dependence on the spatial scale of the outflow-producing system. This observation is in line with the results of Schiro and Neelin (2018), who show that recovery time of the PBL tends to be shorter for isolated convective cells than for MCSs, regardless of the time of the day when the convective activity occurs."*

Pg. 7, line 13 – I wouldn't classify this as a drop; it's more like a "decrease," since it's rather gradual.

**Thank you for pointing that out. We have changed "drop" by "decrease".**

Pg. 7 line 16: instead of "slow", how about "gradual"?

**The word has been changed.**

Insightful discussion in lines 16-22 of pg. 7. I agree with your assessment, since radar reflectivity at 14:57Z does seem to suggest that the cell did not pass directly over the tower.

**Thank you for your comment. In fact, it seems that the cell actually "glanced off" the station site at the time shown in the radar image. It may be speculated that the outflow in the wake of the cell reached the tower site later resulting in the observed gradual decrease in temperature and attendant increase in wind speed.**

Pg. 7, Line 24: I'd be careful about using phrases like "the most organized." It's hard to distinguish organization in the first place (though it's often loosely defined using spatial characteristics). I think classifying it as "organized" is speculative as it is, since you mention that the spatial scale is somewhere in between "isolated" and MCS. Instead, maybe you could classify it as the "system with the largest convective core"?

**We agree with the reviewer's point. Deep convection organization classification is indeed difficult, especially in situations lacking significant vertical wind shear, characteristic of barotropic atmospheric environments. As a result, we incorporated the reviewer's suggestion and change the term "the most organized" to "system with the largest convective core", as it is more appropriate.**

Fig. 3d – Why do you think the 40 m spikes are so much larger (and the data generally noisier) than at 14 and 55 m? Also, where is the rest of the data? Does missing data suggest data quality issues for this sample?

**In Fig. 3d, the data at 40 m had, indeed, quality issues between 3:30 and 5:00 and it has been removed from Figs. 3, 4 and 7. The 80-m data is not available for this event and the 22 m has been added to the Figure.**

Heat flux measurements and discussion: I can't comment too much on the reliability of these data, but I don't doubt that there are noteworthy data concerns here (especially given the really large magnitudes observed in certain instances). At the very least, I think a discussion of the strengths and limitations of using these data during pre-storm and precipitating conditions is warranted in these sections.

**This is a valid concern. Following the suggestion of reviewer #1, we analyze TKE spectra and heat flux cospectra for the 4 different portions of events 1 and 2: before the gust front (I); the period of upward heat flux that marks the gust front arrival (II); the period of large downward heat flux that corresponds to enhanced storm-generated turbulence (III) and the wake period after the event (IV). We also analyze the raw turbulent velocity and temperature data from events 1 and 2 and the precipitation evolution along each event. All plots have been included as supplementary material and a brief discussion referring to them has been included to the manuscript.**

Please explicitly define TKE and how it is computed.

**TKE is computed as:**

$$\mathbf{TKE} = \frac{1}{2}(\acute{u}'^2 + \acute{v}'^2 + \acute{w}'^2),$$

**where:**

**$u'$, $v'$, and $w'$ are turbulent fluctuations relative to the 1-min Reynolds averaged $x$, $y$, and $z$ wind components, respectively, calculated as:**

**$u' = u - \acute{u}$**

**$v' = v - \acute{v}$**

**$w' = w - \acute{w}$,**

**where $u$, $v$, and $w$ represent total (non-averaged) wind components. Overbars indicate Reynolds-averaged quantities.**

**We have included in line 21 (pg. 10) the definition of TKE presented above for clarification.**

References:

Kruijt, B., Malhi, Y., Lloyd, J., Nobre, A.D., Miranda, A.C., Pereira, M.G.P., Culf, A., Grace, J., 2000: Turbulence statistics above and within two Amazon rain forest canopies. Boundary-Layer Meteorol 94:297–331

Saxen, T. R. and S. A. Rutledge, 1998: Surface fluxes and boundary layer recovery in TOGA COARE: Sensitivity to convective organization. Journal of the Atmospheric Sciences, 55, 2763–2781.

Schiro, K. A. and J. D. Neelin, 2018: Tropical Continental Downdraft Characteristics: Mesoscale Systems versus Unorganized Convection. Atmospheric Chemistry and Physics, 18, 1997-2010.

Wang, D., S. E. Giangrande, K. A. Schiro, M. P. Jensen, and R. A. Houze, 2019: The Characteristics of Tropical and Midlatitude Mesoscale Convective Systems as Revealed by Radar Wind Profilers. Journal of Geophysical Research: Atmospheres, 124(8), 4601-4619.

Viswanadham, Y., Molion, L. C. B., Manzi, A. O., Sá, L. D. A., Filho, V. P. S., André, R. G. B., Nogueira, J. L. M., and Santos, R. C., 1990: Micrometeorological measurements in Amazon forest during GTE/ABLE 2A mission. J. Geophys. Res., 95(D9),13669–13682.

---

## Author Comment (AC4) · 17 Oct 2019

*Supplement to*

**Planetary boundary layer evolution over the Amazon rain forest in episodes of deep moist convection at ATTO**

Maurício I. Oliveira[1,a], Otávio C. Acevedo[1], Matthias Sörgel[2], Ernani L. Nascimento[1], Antonio O. Manzi[3], Pablo E. S. Oliveira[1], Daiane V. Brondani[1], Anywhere Tsokankunku[2], and Meinrat O. Andreae[2,4]

[1]Universidade Federal Santa Maria, Departamento de Física, Santa Maria, RS, Brazil
[a]now at: Center for Analysis and Prediction of Storms, and School of Meteorology, University of Oklahoma, Norman, Oklahoma.
[2]Max Planck Institute for Chemistry, P.O. Box 3060, 55020, Mainz, Germany
[3]Instituto Nacional de Pesquisas da Amazônia (INPA), Clima e Ambiente (CLIAMB), Av. André Araújo 2936, Manaus-AM, CEP 69083-000, Brazil
[4]Scripps Institution of Oceanography, University of California San Diego, La Jolla, USA

In the main manuscript, high-frequency, multi-level measurements performed at a 80-m tall tower of the Amazon Tall Tower Observatory (ATTO) are analyzed during the passage of outflows generated by deep moist convection. In order to give the readers confidence in the results, we included in this supplementary material figures of precipitation evolution for all events (Fig. S1), and of raw wind velocity and temperature data for events 1 and 2 (Figs S2 and S5 at all levels). Only events 1 and 2 have been chosen because we could easily identify four different parts in these events from the analysis of sensible heat fluxes: before the gust front (I); the period of upward heat flux that marks the gust front arrival (II); the period of large downward heat flux that corresponds to enhanced storm-generated turbulence (III) and the wake period after the event (IV). Besides, following the suggestion of reviewer #1, a spectral analysis has also been done. For each different portion (I, II, III and IV) of events 1 and 2, we analyzed multiresolution TKE spectra (Figs. S3 and S6) and heat flux cospectra (Figs. S4 and S7) at all levels. Further information regarding multiresolution decomposition can be found at Howell and Mahrt (1997), Vickers and Mahrt (2003) and Voronovich and Kiely (2007).

The main remarks are:

- Precipitation was never large. Total precipitation along the entire duration of the events was 2.3, 1.0, 5.3 and 1.5 for events 1 to 4 respectively;
- Reviewer #1 pointed that "*there may be an issue with vibrations of sensor mounts and tower that affects measurements during storms*". However, TKE spectra and heat flux cospectra are, in all cases, well-organized, tending to zero in the high-frequency limit, indicating that there is reduced levels of noise. Besides, the upward or downward fluxes happen over the entire range of turbulence scales, being well organized vertically as well. They also show that the 1-min time window captures the majority of the turbulent fluctuations. It gives us a high degree of confidence in our dataset.
- The raw velocity and temperature turbulent data from events 1 and 2 indicate the absence of spikes and random fluctuations.

[Figure]

Fig. S1. 1-minute and total precipitation for each event.

[Figure]

Fig. S2. Time series of the velocity components and temperature along event 1.

[Figure]

Fig. S3. Multiresolution TKE spectra for the 4 periods of event 1.

[Figure]

Fig. S4. Multiresolution heat flux cospectra for the 4 periods of event 1.

[Figure]

Fig. S5. Time series of the velocity components and temperature along event 2.

[Figure]

Fig. S6. Multiresolution TKE spectra for the 4 periods of event 2.

[Figure]

Fig. S7. Multiresolution heat flux cospectra for the 4 periods of event 2.

REFERENCES:

HOWELL, J. F.; MAHRT, L. Multiresolution flux decomposition. Boundary-Layer Meteorology, v. 83, n. 1, p. 117–137, 1997. ISSN 1573-1472.

VICKERS, D.; MAHRT, L. The cospectral gap and turbulent flux calculations. Journal of atmospheric and oceanic technology, v. 20, n. 5, p. 660–672, 2003.

VORONOVICH, V.; KIELY, G. On the gap in the spectra of surface-layer atmospheric turbulence. Boundary-Layer Meteorology, v. 122, p. 67–83, 2007.